# Skill of large-scale seasonal drought impact forecasts

Samuel J. Sutanto[1], Melati van der Weert[1], Veit Blauhut2[2], and Henny A. J. Van Lanen[1]

[1]Hydrology and Quantitative Water Management Group, Environmental Sciences Department, Wageningen University and Research, Droevendaalsesteeg 3a, 6708PB, Wageningen, the Netherlands
[2]Hydrological Environmental Systems, University of Freiburg, Fahnenbergplatz, D-79098, Freiburg, Germany

**Correspondence:** Samuel Jonson Sutanto (samuel.sutanto@wur.nl)

**Abstract.** Forecasting drought impacts is still missing in drought early warning systems (DEWS) that presently do not go beyond hazard forecasting. Therefore, we developed drought impact functions using machine learning approaches (Logistic Regression and Random Forest) to predict drought impacts with lead-times up to 7 months ahead. The observed and forecasted hydrometeorological drought hazards, such as the Standardized Precipitation Index (SPI), Standardized Precipitation Evaporation Index (SPEI), and Standardized Runoff Index (SRI), were obtained from the ANYWHERE DEWS. Reported drought impact data, taken from the European Drought Impact Inventory (EDII), were used to develop and validate drought impact functions. The skill of the drought impact functions to forecast drought impacts was evaluated using the Brier Skill Score and Relative Operating Characteristic metrics for 5 Cases representing different spatial aggregation and lumping of impacted sectors. Result shows that hydrological drought hazard represented by SRI has higher skill than meteorological drought represented by SPI and SPEI. For German regions, impact functions developed using Random Forest indicate a higher discriminative ability to forecast drought impacts than Logistic Regression. Moreover, skill is higher for Cases with higher spatial resolution and less-lumped impacted sectors (Cases 4 and 5), with considerable skill up to 3-4 months ahead. The forecasting skill of drought impacts using machine learning greatly depends on the availability of impact data. This study demonstrated that the drought impact functions could not be developed for certain regions and impacted sectors, owing to the lack of reported impacts.

## 1 Introduction

Drought is a creeping phenomenon that slowly covers and affects extensive areas (Wilhite, 2000; Tallaksen and Van Lanen, 2004; Mishra and Singh, 2010; van Loon, 2015). It can last from months to years and can have multi-faceted impacts on, for example, crop production, drinking water supply, electricity production, waterborne transportation, and recreation (Wilhite, 2000; UNDP, 2006; EEA, 2007; Stahl et al., 2016). In terms of the number of affected people and economic losses, drought is categorized as one of the most damaging natural hazards (Wilhite and Vanyarkho, 2000; EEA, 2012). Worldwide, 2 billion people were affected and more than 10 million people passed away, due to the impacts of drought between 1900 and 2010 (Georgi et al., 2012). In the period 1976-2006, 15% of the European territory and 17% of the European population was affected by drought on an average annual basis and it was estimated that the total damage amounted to $> 100$ billion Euro (EEA, 2007).

Actions, such as the development of drought monitoring and early warning systems, as it is highlighted in the UN Sendai Framework for disaster risk reduction (Poljanšek et al., 2017), have been taken to reduce the impacts of drought. However, present drought monitoring and early warning systems, only provide the end-users with information of present and forecasted drought hazard conditions, mainly derived from the hydro-meteorological drought indices (Sepulcre-Canto et al., 2012; Pozzi et al., 2013; Sheffield et al., 2014; Sutanto et al., 2019a). Information on forecasted drought conditions, expressed by a drought hazard index, does not directly translate the hazard into impacts. Thus, the forecasting of drought impacts is still missing in drought early warning systems. This may be due to finding information on drought impacts is quite challenging, as drought occurs in different hydrological domains and has mainly non-structural consequences (Lackstrom et al., 2013; Stahl et al., 2016; Kreibich et al., 2019). Nevertheless, the likelihood of impact occurrence, which is a result of the hazard, exposure, and vulnerability, is essential to stakeholders in order to manage drought events. Relating drought hazard indices with their impacts is beneficial for decision-making and risk management (Hayes et al., 2011). For water managers, stakeholders, and policymakers, there is a vital need to be informed about forecasted impacts of drought events, rather than only about the drought hazard. Thus, adaptation actions can be prepared and taken well in advance, to reduce the impact of drought.

As alternative to conventional bottom-up approach that links potential risks/impacts to hazard, exposure and vulnerability (e.g. Carrão et al. (2016); Naumann et al. (2014, 2015)), some studies have been conducted to develop a link between drought impacts and the underlying hazard (top-down approach). Gudmundsson et al. (2014) were, among others, the first to study this relationship using the Standardized Precipitation Index (SPI, McKee et al. (1993)) and the European Fire Database (EDF, Camia et al. (2014)). This study in southern Europe showed that there is a significant relationship between the probability of above normal wildfire incidents and meteorological drought hazards. The development of the European Drought Impact Inventory (EDII; Stahl et al. (2016)), a unique collection of categorized, temporally, and spatially assigned drought impact information, offers novel possibility to evaluate drought impacts. Using the EDII database, Blauhut et al. (2015) assessed the link between drought hazard and various impacted sectors across Europe on an annual scale, such as Agriculture and Livestock farming, Energy and Industry, Pubic Water Supply, and Water Quality. Stagge et al. (2015) enhanced this link by accounting for seasonality, incorporating inter-annual trends, and allowing for non-linear effects. In a joint publication, Blauhut et al. (2016) developed a method to test currently monitored drought hazard indices and vulnerability information on their ability to model the likelihood of impact occurrence. All the studies described above used similar methods to derive predictor-impact relations, which were based on the machine learning method, Logistic Regression (LR). In addition, Bachmair et al. (2017) assessed further machine learning approaches for deriving the link between drought hazard and their impacts, such as zero-altered negative binomial regression, and Random Forest (RF) models.

The aforementioned studies in general used historical observational data to derive the meteorological drought indices (SPI and Standardized Precipitation Evaporation Index, SPEI Vicente-Serrano et al. (2010)) to construct drought impact functions. The results were promising and hence it was hypothesized that these also could be implemented in the forecast mode. Sutanto et al. (2019b), therefore, have conducted a first study that explored the possibility of forecasting the drought impacts using hydro-meteorological drought indices, incl. the SPI, SPEI and the Standardized Runoff Index (SRI, Shukla and Wood (2008)) and the EDII database with data from Germany. They only used the RF method to forecast drought impacts for four different

lumped impacted sectors (i.e. impact groups). Their study shows that, in general, forecasting drought impacts is feasible, with skill up to a few months ahead depending on the number of reported impacts and drought duration. However, the skill likely depends on the machine learning approach adopted to derive functions that relate drought hazard indices to impacts (i.e. impact functions).

In this study, we enhance the previous studies by evaluating the skill of seasonal drought impact forecasts using the RF and LR approaches for different combinations of spatial aggregations and impact categories (i.e. Cases), which is the novelty of our paper. The RF and LR approaches have proven value in developing drought impact functions using the historical data and were not used yet for drought impact forecasting study using dynamical weather forecasts. One should note that a study conducted by Sutanto et al. (2019b) only used the RF approach and lumped impacted sectors. To evaluate the skill of seasonal drought impact forecasts using the RF and LR approaches. We divided the study into three parts with the following objectives (Fig. 1): 1) evaluation of the skill to forecast drought hazards using re-forecast data from 2002-2010 (box G), which is a first step in the forecasting chain, 2) evaluation of the skill of the drought impact functions based upon the RF and LR approaches using proxy observed data and a split sampling method (box L), and 3) testing the skill of developed drought impact functions based upon the RF and LR approaches to forecast drought impacts using re-forecast data from 2002 to 2010 (box O).

## 2 Data and methods

The flowchart that summarizes the Data and Method section, as well as the objectives of the three parts of the study, is presented in Figure 1. In general, two datasets and two machine learning approaches were used in this study. The hydro-meteorological data (yellow boxes in Fig. 1) were required to calculate the drought hazard indices (SPI-x, SPEI-x and SRI-x). The drought impact data (green box in Fig. 1) were prerequisite to translate the drought hazard into the Likelihood of drought Impact Occurrence (LIO) in the next phase using the binary Logistic Regression (LR) model and the Random Forest (RF) model (blue boxes in Fig. 1). Three forecast skill analyses were carried out to assess 1) the drought hazard forecast skill (objective 1), 2) impact forecasting skill by using observed impact data and a split sampling technique (objective 2), and 3) drought impact forecasting skill by using re-forecast data (objective 3).

### 2.1 The EDII database

The EDII is a joint effort of the EU FP-7 project Drought RSPI (eu-drought.org), which is actually hosted, updated, and maintained by the University of Freiburg (Stahl et al., 2016). The database is a compilation of drought impact reports in Europe, which stem from a variety of sources e.g. governmental reports, scientific publications, media, or questionnaires (box A in Fig. 1). In the EDII, drought impacts are considered as negative environmental, economic, or social effects of the hazard of drought. Hence, drought conditions, such as an anomaly in precipitation, soil moisture, groundwater levels, or runoff, which do not result in a negative consequence, are not considered as an impact in the EDII inventory. Each impact report is assigned to: 1) temporal reference; this can be indicated by month, season or year as a minimum, 2) spatial reference; the location of the reported impacts can be either referred to different levels of geographical regions using the European Union Nomenclature

of Units for Territorial Statistics (NUTS) (Eurostat, 2011), or specified by rivers or lakes, and 3) one of 15 impact categories and a selection of 105 impact types (see Stahl et al. (2016) for detailed information on the EDII database). In this study, drought impact reports from the period 1990-2017 for Germany at the country and county level (NUTS-1 regions, Fig. S1 in the supplementary for the Germany NUTS-1 regions) were used. For each NUTS region and impacted category, binary time-series (0: no drought, 1: drought) of monthly reported drought impacts were derived from the EDII (1990-2017, box H Fig. 1) following Blauhut et al. (2015) and Bachmair et al. (2017).

Since the availability of impact information from the EDII database appears to vary across Europe (Stahl et al., 2016; Sutanto et al., 2019b), we developed drought impact functions for the five different Cases, i.e. combinations of spatial aggregations and impact categories. For Cases 1 and 2, we accumulated the entire country of Germany and for Cases 3, 4, and 5, a higher spatial resolution, i.e. NUTS-1 regions (Fig. S1), was applied. For the Cases 2 and 5, all 15 individual impacted sectors were tested to forecast drought impacts. For the Cases 1 and 3 all sectors were lumped. For Case 4, the EDII impacted sectors were arbitrarily grouped as follows: Group 1 consists of agriculture and livestock farming and forestry, Group 2 consists of Energy and Industry, Transportation, and Public Water Supply, Group 3 consists of Water Quality, Freshwater Ecosystem, and Terrestrial Ecosystem, and Group 4 consists of Wildfire, Air Quality, and Human Health and Public Safety. We grouped the impacts based on their likely relation to specific drought types (meteorological, agricultural, and hydrological drought) and topic (water quality, ecosystems). Group 1 relates to agricultural drought, group 2 relates to streamflow drought (hydrology), group 3 relates to hydrological drought linked to ecosystems, and last, group 4 relates to meteorological drought and heat. These Cases and the selection of the EDII impacts are documented in Table 1. Hence, the binary time-series differ and case specific (the more impact categories and regions merged, the more reported impacts reported). The impacts that were reported as a season were transformed into months of that season (e.g. winter = December, January, and February, spring = March, April, and May, and so on).

## 2.2 Hydro-meteorological dataset

The hydro-meteorological data used in this study were obtained from the European Flood Awareness System (EFAS, Thielen et al. (2009); Pappenberger et al. (2011)) with a spatial resolution of 5 km by 5 km. The EFAS used in the study is based upon a state-of-the-art LISFLOOD rainfall-runoff model (Bartholmes et al., 2008; van der Knijff et al., 2010; Burek et al., 2013) and the European Centre for Medium-Range Weather Forecasts (ECMWF) seasonal forecasts S4 (SEAS4, Smith et al. (2016)). Gridded observed meteorological data, such as precipitation and evapotranspiration were used by the LISFLOOD model to simulate gridded runoff. In EFAS, potential evapotranspiration is calculated through the offline LISVAP pre-processor based on the Penman-Monteith equation (van der Knijff, 2008). The gridded meteorological and runoff data (Simulation Forced with Observed, SFO, hereafter referred as proxy observed data, box B Fig. 1) were used to calculate the observed drought hazards with a time scale of 1990-2017 (box E Fig. 1). Simulated runoff had to be used since gridded runoff cannot be measured. Streamflow observational data in some rivers are available (>250 catchments across Europe) and these were used to calibrate and validate the LISFLOOD hydrological model. LISFLOOD obtained a median Nash Sutcliffe Efficiency (NSE) of 0.57 over the validation period (Feyen and Dankers , 2009; Forzieri et al., 2014).

Meteorological and hydrological re-forecasts (in many cases referred to as hindcasts) were obtained from the ECMWF meteorological forecast SEAS4 and the LISFLOOD model forced with the ECMWF SEAS4, respectively (box C Fig. 1). These re-forecast data were used to calculate the forecasted drought hazards from 2002 to 2010 (box F Fig. 1), using the distribution parameters derived from the observed datasets (box D Fig. 1). Re-forecasts of meteorological and hydrological variables have a lead-time of 215 days (circa 7 months) and consist of 15 ensemble members. Detailed information on the hydro-meteorological dataset from EFAS can be found in Arnal et al. (2018), van Hateren et al. (2019), and Sutanto et al. (2019a). The daily proxy observed and re-forecast data were transferred into monthly time series.

## 2.3 Standardized drought hazard indices

The calculation of hydro-meteorological drought hazards was carried out using the Standardized Precipitation Index (SPI, McKee et al. (1993)), the Standardized Precipitation Evaporation Index (SPEI, Vicente-Serrano et al. (2010)), and the Standardized Runoff Index (SRI, Shukla and Wood (2008)), with accumulation periods of 1, 3, 6, and 12 months (box E and F Fig. 1). The standardized indices were calculated by fitting a probabilistic distribution on monthly observed variables (1990-2017): precipitation for SPI, precipitation minus potential evapotranspiration for SPEI, and runoff for SRI. These input data were averaged for the German counties (NUTS-1) and country before the calculation of the standardized indices was performed. A gamma distribution (SPI and SRI), and three-parameter log-logistic distribution (SPEI) were selected to transform the monthly hydro-meteorological data into 144 distributions for every month of the year, for each drought hazard index, and for each accumulation period (12x3x4, box D Fig. 1). We used the distributions to calculate monthly time series of the three drought hazard indices (SPI, SPEI, SRI) for the observed data (1990-2017) (box E Fig. 1) and re-forecasted data (2002-2010, box F Fig. 1).

## 2.4 Machine learning for modeling drought impacts

LR predicts the probability of a dichotomous variable. This model belongs to the class of Generalized Linear Models (GLMs, Zuur et al. (2009)). The likelihood of impact occurrence (LIO, logit-transformed) is modeled as a linear function of drought hazard predictors as:

$$log\frac{LIO}{1-LIO} = \alpha + \sum_i \beta_i x_i \tag{1}$$

The left side of the equation represents the logit transformation of LIO. The model parameter $\alpha$ and $\beta$ are intercept and coefficient, respectively, that were estimated using standard regression techniques within the framework of GLM (Peng et al., 2002). $x$ is predictor variables e.g., month, year, SPI-x, SPEI-x, and SRI-x, with different accumulation periods, and $i$ is the number of predictor variables. The months of impact occurrence (e.g. January, February, and so on until December) were included to account for seasonality, as some drought impacts are connected to particular seasons. The years of impact occurrence were included (e.g., 2003, 2010, 2015, and so on) as a predictor since drought impacts gradually have been better documented throughout the years. To find the best LR models for LIO (box J, M Fig. 1), we applied three steps following

Blauhut et al. (2016) as follow. 1) A selection from the initial predictors (SPI-x, SPEI-x, SRI-x, month, and year) was tested for their single ability to predict the impact. Insignificant predictors indicated with a probability value ($p$) larger than 0.05 were omitted, and the remaining predictors were tested for collinearity (step 2). 2) Each combination of 2 independent variables was

tested for their collinearity. For predictors, which were highly correlated to each other ($p > 0.7$), the best was kept. Thereafter 3), the forward and backward stepwise selection was used to find the best combination of predictors defined by the Akaike Information Criterion (AIC). This analysis was performed using the "MASS" package in Rstudio (Ripley et al., 2019). In this procedure, the model started with no independent variables. Sequentially, the most significant independent variable was added to the model and was examined for the model improvement. The independent variable was removed from the model if it did not

increase the model performance. This procedure was performed until all selected independent variables from step 1 and 2 were analyzed. The selection procedures of the best predictors mean that not all predictors are used in the LR method. To estimate the contribution of each variable to the model, the absolute value of the t-statistic for each model parameter was assessed using the "Caret" package in Rstudio (Kuhn, 2019).

Random Forest (RF) is a Machine Learning algorithm that is powerful to develop a predictive model (Breiman, 2001).

The system creates a multitude of random independent decision trees as an ensemble to reduce the chance of overfitting and sensitivity to training data configuration (in our case 2000 trees). Each tree is constructed based on a bootstrapped fixed size of a subsample of the data and predictions are made by the mean prediction of the individual trees. The RF method was performed using the "RandomForest" package in Rstudio (Liaw and Wiener, 2002). We used all the predictor variables in the RF model (box E Fig. 1) and the response variables consist of the binary time-series of reported drought impacts from the EDII (box H

Fig. 1). We did not do cross validation our model, instead, we used the "Caret" feature to identify the drought hazard indices best linked to impact occurrence (Kuhn, 2019). The Caret uses the prediction accuracy on the out-of-bag (OOB) portion for both the full model and after permuting each predictor variable. The differences between the two models are then averaged over all trees, and normalized by the standard error. By doing this, the performance of the model was tested. Likelihood of Impact Occurrence (LIO) was estimated by calculating the probability of the number of trees that indicated the impact for every

German NUTS region. We trained the machine learning models (LR and RF) using the data from 1990 to 2015 for objective 2 (box J Fig. 1). For objective 3, we used all observations from 1990 to 2017 for model trainee (box M Fig. 1). Bachmair et al. (2016), Bachmair et al. (2017), and (Sutanto et al., 2019b) provide a detailed explanation of the use of RF to develop drought impact forecasting functions.

## 2.5    Forecasting skill score

The skill of forecasted drought hazard indices and drought impacts was evaluated using common skill score metrics, such as the Brier Skill Score (BSS, Brier (1950)) and Relative Operating Characteristic (ROC, Mason (1982)). The BSS can be used to measure both the reliability and sharpness of forecasts (Trambauer et al., 2015; Arnal et al., 2018). For the BSS, both the Brier Score for the climatology (BSclim) and the Brier Score for the forecast (BSf) are required. The Brier Score (BS) measures the mean squared difference between the predicted probability and the verification sample. For BSclim, a threshold of -0.5 was

chosen to give a good balance between capturing either too many minor droughts or too few drought events (Trambauer et al.,

2015). The standardized index of -0.5 stems from a normal distribution, and therefore the value for $p$ is 0.3085 for every month (van Hateren et al., 2019). The BSS measures the improvement of the probabilistic forecasts relative to climatology. It has a range of [1,-∞] where 1 indicates the highest skill, and values below 0 indicate that the probabilistic forecast does not show additional predictive performance against the climatology.

The ROC curve was used as a criterion to measure the discriminate ability between two outcomes (Mason and Graham, 1999). The ROC curve gives the relation between the true positive rate (sensitivity) and the false positive rate (specificity). The Area Under the Curve (AUC) was calculated to measure the accuracy of the forecast. The AUC has a range from 0 to 1, with a perfect score of 1. AUC=0.5 is as good as flipping a coin and AUC<0.5 is unskillful.

## 3 Results

### 3.1 Skill of drought hazard forecasts

Skill of probabilistic drought hazard forecasts (SPI-x, SPEI-x, and SRI-x) for our example, Germany, using median of the ensemble of re-forecast data from 2002 to 2010 is presented in Figure 2. The skill was measured using the BSS and ROC skill scores and presented for forecasts done in each season (colored lines) and for the annual time scale (black line). Figure 2 shows that drought indices with longer accumulation periods perform better than the ones with short accumulation periods. For example, SPI-1 forecasts using the BSS show no skill for all seasons except for forecast done in autumn for a lead-time (LT) of 1 month. On the other hand, the skill much improves for longer accumulation periods, i.e. SPI-12, with BSS values around 0.7 for 1-month LT. The lowest skill is found for forecasts done in summer for all accumulation periods. As expected, the skill also decreases for higher LTs.

The predictive skill of drought hazard forecasts varies between the drought indices. In general, SPI-x and SPEI-x have similar predictive skill for long accumulation periods (e.g., x=6 and 12 months). However, the SPEI-x index shows higher skill than the SPI-x for short accumulation periods. The SPI-3 forecasted during summer shows a BSS value of 0.75 while the SPEI-3 shows a BSS value of 0.85. In general, the SPEI drought forecasts have slightly higher skill than the SPI. The source of predictability of SPEI comes from the higher temperature predictability due to the North Atlantic Oscillation (NAO) in Europe than precipitation (Vitart, 2014). Temperature is one of the weather variables that controls potential evapotranspiration. The hydrological drought index, SRI-x, predicts drought better than the meteorological drought indices, e.g., SPI-x and SPEI-x. This is illustrated in the BSS and ROC figures. The BSS for SRI-1, for example, shows that there is a predictive value up to a LT of 2 months, which indicates a similar skill than SPI-3 and SPEI-3. For drought indices with longer accumulation periods, SRI-x performs better for almost all LTs than SPI-x and SPEI-x. The comparison between the BSS and ROC skill score learns that drought forecasts using the ROC method produces a higher score. This is clearly seen for droughts forecasts with higher accumulation periods and LTs (Fig. 2).

## 3.2 Skill of drought impact forecasts using observed data

To test drought impact function performance for the five different Cases (box L Fig. 1 and Table 1), we developed these functions using the observed data from 1990 to 2015 (Box J Fig, 1). The predictive power of the model is assessed by comparing with observed impacts from the last 2 years (2016 to 2017, split sampling technique). Please note, no re-forecasted data were used in this analysis. A summary of the model performance using the ROC skill score for the different Cases is presented as boxplots in Figure 3. Please keep in mind that Case 1 only covers one impact function, which is developed using all impact data lumped, irrespective of the impacted sector, for Germany (all NUTS-1 regions). Hence, Case 1 is presented as a single line in Figure 3. For each boxplot, the median of the ROC values obtained from all models is also presented as a single line.

Figure 3 shows that, overall, the models have skills in distinguishing impact and no impact. This is indicated by the median for all five Cases above 0.5. The analysis of the medians between the two machine learning approaches shows a large difference in skill score for Case 1. The model's skill score for LR (ROC=0.53) is substantially lower than RF (ROC=0.7). However, both methods show similar skills for the other Cases, which have a substantially lower number of impact reports. Cases 2 and 3 show lower performance than Cases 4 and 5. Variation within the Cases seems more pronounced in LR than in RF.

## 3.3 Skill of drought impact forecasts using re-forecast data

After testing the drought impact functions performance using a split sampling technique (Section 3.2), we re-developed drought impact functions using all observed datasets from 1990 to 2017 (Box M Fig, 1). These models were built as if these would be implemented in a drought early warning system. Another reason to use the full dataset is that we would like to train the models with as much data as possible. After developing the functions, these were used to forecast the drought impacts (Box N Fig, 1) using the re-forecast data from 2002-2010 (Box F Fig, 1). A summary of the skill of drought impact functions to forecast drought impacts (box O Fig. 1) using the ROC skill score is presented in Figure 4.

In Figure 4, we divide the analysis of drought impact forecasting skill into short LTs (1-3 months) and long LTs (4-7 months). Obviously, forecasting drought impacts for shorter LTs produce a higher skill than longer LTs. This is true for all Cases and the two modeling approaches. Figure 4 also shows that the RF performs better than the LR for all LTs. If only the median value is considered, models developed using the LR method have skill only for Cases 3, 4, and 5 and only for short LTs. For long LTs, all the LR models show no skill. For RF, however, skill is obtained for both short and long LTs and for all Cases. Variation within Cases seems to be smaller for RF than LR and the differences are larger for longer LTs than for shorter ones. The skill of drought impact forecasts using LR for short LTs (Cases 3, 4, and 5) varies around 0.74 whilst the skill using RF fluctuates around 0.85.

As discussed in the previous paragraph, the highest skill is achieved using the RF method for Cases 4 and 5. Therefore, the skill of drought impact forecasts developed using RF for Case 4 is presented in Figure 5, for each German NUTS-1 region and different impact groups. We only present the skill of drought impact forecasts for Case 4 because the number of drought impact functions that can be developed for Case 5 is limited. This is due to the large differences in the number of reported impacts for each impacted sector between the German NUTS-1 regions (Stahl et al., 2016; Sutanto et al., 2019b). For example, for Case 5,

drought impacts on Agriculture and Livestock Farming or Forestry can be forecasted in BW and BV regions only, respectively (Figure not shown). However, if we combine these two impacted sectors into one group (Group 1, Case 4), impact functions can be developed for regions BW, BV, RP, and LS (Fig. 5).

Figure 5 shows that even if we combine several impacted sectors into four different impact groups, drought impact functions can be developed only for some regions in Germany due to the aforementioned data limitation and regional characteristics. For example, HH and HB are mainly cities, hence, there are no reported agriculture and livestock farming impact. Based on the BSS skill score, good drought impact forecasting skill is achieved only for impact Group 2 in BV. However, the ROC skill score shows a good skill for all impact groups for short LTs. In general, the skill of drought impact forecasts using BSS is fair for short LTs and poor for longer LTs implying that for shorter LTs a reasonable distinction can be made between impact and no impact. This is especially the case for situations where the observed data show prolonged periods with drought impacts (Sutanto et al., 2019b).

## 4 Discussion

The use of standardized drought indices i.e., SPI, SPEI, and SRI, require preferably 30 years record (McKee et al., 1993; Vicente-Serrano et al., 2010; Shukla and Wood, 2008). In our study, we could only use 28 years of observational record (proxy) from 1990 to 2017 (Section 2.2). EFAS data before 1990 are not available and we only had data up to 2017. The length of the observational data might have some implications on the calculation of parameters of the monthly distributions, which affects the calculation of the drought severity index. For example, if the records (< 30 years) do not include extreme low/high events, then the calculation of drought indices will overestimate these low/high severities due to lack of low/high data in the head/tail of the normal distribution. In our study, 2 years of missing data in the record do not significantly influence our results for following reasons: 1) years 1988 and 1989 were not recorded as extreme drought years, and 2) we included drought years in parts of Europe, e.g. 1991-1992, 1995, 1996-1997, 2003, 2006, 2008, and 2015 (Spinoni et al., 2015).

Analysis of model performance using the split sampling method (Fig. 3) indicates that both machine learning approaches (LR and RF) seem to have similar descriptive power. However, for drought impact models evaluated using re-forecast data, RF shows better performance, denoted by higher ROC scores (Fig. 4). For evaluation using re-forecast data, the drought impact models were trained using all observed data from 1990 to 2017. This implies that RF seems to have a longer memory than LR, meaning that RF keeps memory of the trained data, which were used to build the drought impact functions. Moreover, the number of predictors used in RF and LR differs because of the selection processes in the LR method, as done in previous studies (Stagge et al., 2015; Blauhut et al., 2015), whereas in RF, we used all predictors (Bachmair et al., 2016).

Figures S2 and S3 provide the importance of predictors in LR and RF, respectively. Figure S2 clearly shows that in the LR method only some of the predictors were used, whereas, in the RF method, all the predictors were used (Fig. S3). For example, in BV for impact Group 2, LG only used SPI-3, SPEI-12, and year as the predictors, with SPEI-12 as the most important predictor followed by SPI-3 and year. Using the RF method to predict drought impact Group 2 in BV, the most important predictors are SPEI-6, SRI-6, and year, while the less important predictors are SPI-1, SPI-12, and SPEI-1. In this study, we

did not use the same numbers of predictor for both LR and RF, instead, we followed the same approach as in previous studies (Stagge et al., 2015; Blauhut et al., 2015; Bachmair et al., 2016, 2017). Whether the skill of drought impact functions developed using LR (RF) would improve (decrease), if we trained the models using the same number of predictors, is subject to future study.

It is also worth to mention that the RF has several advantages relative to the LR in this study because: (i) it can handle non-linear relationships among variables, (ii) it is robust to noise, (iii) the bootstrap sampling provides a way to account for the uncertainty in the impact data, and (iv) it has higher the true positive rate (Breiman, 2001; Hastie et al., 2009; Evans et al., 2011; Trigila et al., 2015; Kirasich et al., 2018; Fung et al., 2019). Among some advantages of the RF approach mentioned above, the ability of RF to handle non-linearity is of the utmost importance in drought study. The drought severities obtained from the standardized indices (i.e. SPI, SPEI, and SRI), i.e. input data, were derived from functions, which were developed by fitting the gamma distribution that later were transformed into normal distribution (Section 2.3). This indicates that the drought severities, as input data for RF and LR are not linear, which is conjectured for better skill of RF than LR. Furthermore, the RF has the potential to handle information on the severity of the impact, which is beneficial for future decision making (Breiman, 2001; Bachmair et al., 2016, 2017). Our results support previous drought studies, which concluded that RF performs better than any other machine learning methods, such as LR, Boosted Regression Trees, Cubist, decision trees, and Hurdle (Park et al., 2015; Bachmair et al., 2017; Rhee and Im, 2017).

Machine learning is a data-driven approach, in which the model performance is strongly related to the quality and quantity of the underlying data (Solomatine and Shrestha, 2009; Elshorbagy et al., 2010; Mount et al., 2016). Drought impact functions could not be developed in HH, HB, ST, TH, and SL for any Case because there are less than 20 months with reported impacts in the period 1990 to 2017 (Fig. S4). Many reports were collected in the south of Germany, while only a few reports were collected in the north of Germany. For example, many reports were found in BV and BW (>120 months) during the study period of 1990-2017. However, less than 20 months of reports were collected in SH, HH, HB, ST, BE, TH, and SL (red line Fig. S4). The total number of reported impacts in the whole of Germany from 1990 to 2017 is ~1160. Combining all impact data from all impacted sectors for a NUTS-1 region helps to build a drought impact function in a region that has a limited number of reported impacts. However, if impacts on all sectors are lumped (Case 3), then obviously, the model only provides total impact occurrence (impact or no impact) without any information on impacted sectors. For example, a drought impact model can be built in the SH region, which has a total number of reported impacts from around 20 coming from eight different impacted sectors. In this case, the model can predict the impact occurrence relatively well, with a ROC value of 0.8 for LT=1 (Figure not shown). When we break down all impacts for a NUTS-1 region into impact groups (Case 4), meaning less reported impacts for each group, a drought impact model cannot be made in this region (Fig. 5). Thus, a good balance between the number of groups and reported impacts in each group by considering similar impact types and causes should be taken into account when developing drought impact models. The forecasting skill assessment was carried out using the BSS and ROC approaches for drought hazard and drought impact forecasts. Our results show that, in general, the BSS skill score produces a lower skill than ROC. The ability of the BSS and ROC to distinguish between good and fail forecasts is limited by the number of years in the re-forecast period (Kumar, 2009) and ensemble members (Scaife et al., 2014; Dunstone et al., 2016). The BSS

assesses all joint distributions of forecasts and observations based on the ensemble members in probability binary time series. Therefore, BSS is subject to large sampling variability and more sensitive to the details of the datasets (Hamill and Juras, 2006). Conversely, the ROC skill score uses the probability thresholds of the ensemble members. Thus, it may subject to be noisier but it is insensitive to the particular probabilities themselves and insensitive to bias (Bett et al., 2018). The BSS is also sensitive to rare events, such as extreme wet and dry conditions. Therefore, large sample sizes are required to stabilize the BSS when one evaluates rare events. The reliability of the BSS values of the re-forecasted drought impacts is uncertain as these events are subjected to a more skewed distribution, which is not the case for ROC. In the ROC analysis, the success rate in predicting no impact is not part of the goodness-of-fit measure. Therefore, the ROC is not sensitive to biases and does not include reliability. The downside, however, is that it ignores the predicted probability values and the goodness-of-fit of the model.

## 5    Conclusions

This study evaluates the skills of both drought hazard and impact forecasts, which were developed using observed and re-forecasted hydro-meteorological data, and drought impact reports (EDII). Empirical drought impact functions were developed by utilizing Logistic Regression (LR) and Random Forest (RF) machine learning approaches to link drought hazards to drought impacts.

Our results show that re-forecasted drought hazard indices with longer accumulation periods have higher skill than those with short periods. Longer accumulation periods consist of a longer series of observation data relative to the lead-time, which is a caveat for forecasting skill assessment. The higher proportion of observed data will artificially inflate forecast scores. Likely, forecast skill of drought indices with longer accumulation periods also is positively affected by the lower fluctuation/noise in the index time series compared to shorter accumulation periods. Furthermore, higher prediction skill is obtained for hydrological drought hazard indices than for meteorological indices for the same accumulation period, which is associated with the memory represented in initial hydrological conditions and storage in the hydrological system (Sutanto et al., 2020).

The descriptive power of Logistic Regression (LR) and Random Forest (RF) to forecast drought impacts was assessed for multiple Cases. Both data-driven models show similar skills for predicting drought impact occurrence when compared with observed impact data from 2016-2017. However, when the drought impact functions that were trained using all observed impact data and drought hazard indices (1990-2017), were used to predict drought impacts based upon re-forecast data (2002-2010), RF shows higher discriminative ability than LR. Our findings also reveal that models with higher spatial resolution and less-lumped impacted sectors (Cases 4 and 5) have higher predictive skill than the ones for more generic Cases (Cases 1, 2, and 3). For the former Cases, drought impacts can be predicted up to 3-4 months ahead with considerable skill.

The forecasting skill of drought impacts greatly depends on the availability of impact data. Data-driven models require sufficient reported impact data for training the model. This study demonstrated that the drought impact functions could not be developed for certain regions and impacted sectors, owing to the lack of reported impacts. Although higher skill was gained for Case 5 than Case 4, less drought impact functions could be developed for Case 5. This is a reason that we are in favor of lumping impact data into several impact groups (Case 4). In this study, we used all drought hazard indices with all accumulation

periods (SPI-x, SPEI-x, and SRI-x) to develop drought impact functions using RF. A further study to develop drought impact functions based on pre-defined drought indices with certain accumulation periods, which are linked to specific impacts, is suggested. For example, drought indices with long accumulation periods are used to forecast hydrological drought impacts and vice versa for meteorological drought impacts.

*Data availability.* The EFAS data are accessible under a COPERNICUS open data license (https://doi.org/10.24381/cds.e3458969). Other
data generated and/or analyzed during this study are freely available in the 4TU Center for Research Data (doi:10.4121/uuid:4c6f7f0f-e402-4b6c-a2a2-711bcd224e1f).

*Author contributions.* All authors conceived and implemented the research. Data analyses, model output analyses, and all figures have been performed by M.V.D.W and S.J.S. S.J.S and H.A.J.V.L wrote the initial version of the paper. V.B contributed to interpreting the results, discussion, and improving the paper.

*Competing interests.* The authors declare no competing interests.

*Acknowledgements.* The research was supported by the ANYWHERE project (Grant Agreement No.: 700099), which is funded within EU's Horizon 2020 research and innovation program (www.anywhere-h2020.eu). The hydro-meteorological output came from the EFAS Computational Center, which is part of the Copernicus Emergency Management Service (EMS) and Early Warning Systems (EWS). Finally, we acknowledge the Ministry of Science, Research and the Arts of the State of Baden-Württemberg for financing the DRIeR-project, which
maintains EDII. This research is part of the Wageningen Institute for Environment and Climate Research (WIMEK-SENSE) and it supports the work of UNESCO EURO FRIEND-Water program and the IAHS Panta Rhei project Drought in the Anthropocene.

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

525

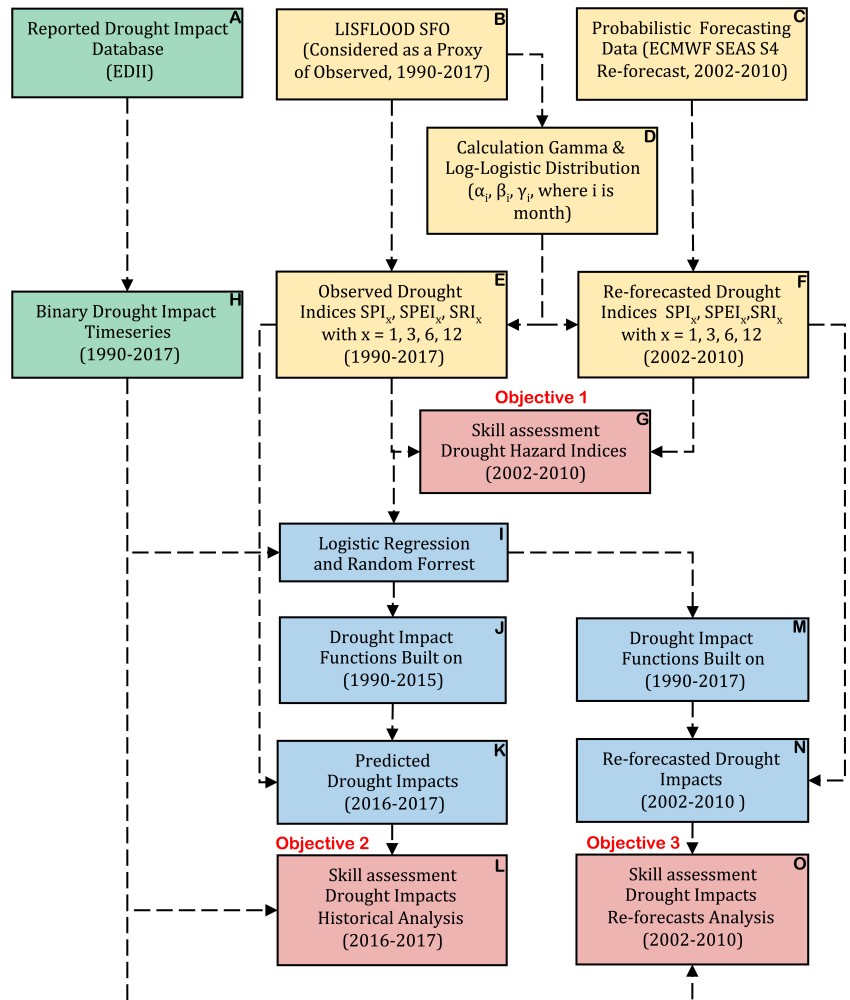

**Figure 1.** Workflow diagram, including colored boxes to illustrate the main components (drought hazard = yellow, drought impact = green, linking impact to hazard = blue, and validation of forecasting skill = red), as well as the objectives of the three parts of the study.

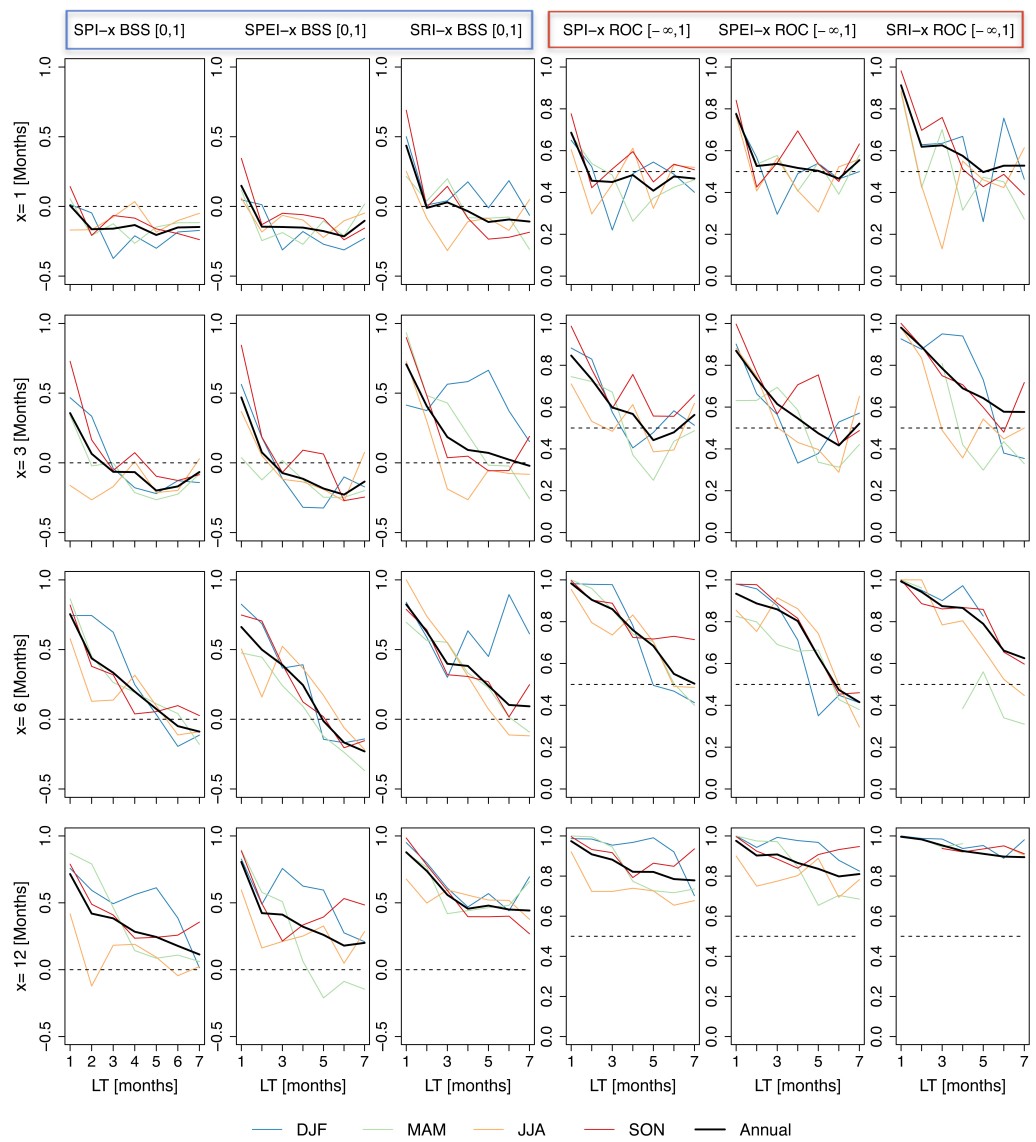

**Figure 2.** Skill scores of re-forecasted drought hazards indices (SPI-x, SPEI-x, and SRI-x) for 7-month lead-times (LTs) in Germany for the period 2002-2010 (median of 15 ensemble members). The 4 different rows of graphs represent 4 different accumulation periods (x=1, 3, 6, and 12 months). The black dashed line indicates the threshold (BSS=0, ROC=0.5). The first 3 columns show the BSS for the 3 different drought indices (indicated by a blue box at top). The last 3 columns are for ROC (indicated by a red box at top).

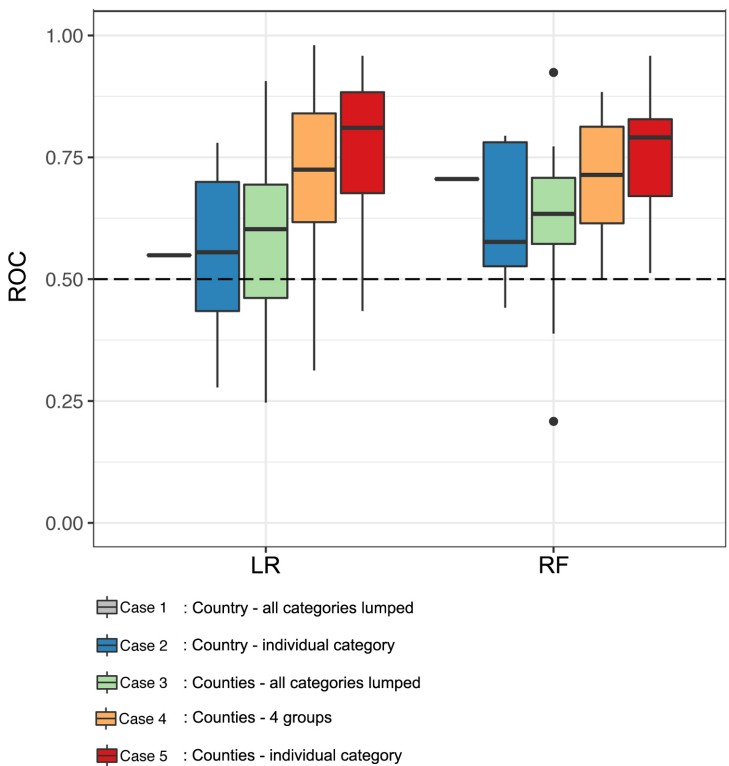

**Figure 3.** Model performance for the five Cases using Logistic Regression (LR) and Random Forest (RF) approaches to predict drought impacts for the period 2016-2017. The boxes indicate the spread of ROC values for drought impact functions (15 ensemble members). The black dashed line indicates the threshold (ROC=0.5). All values below this line models have no discriminative capacity to distinguish impact and no impact. The dots stand for models outliers.

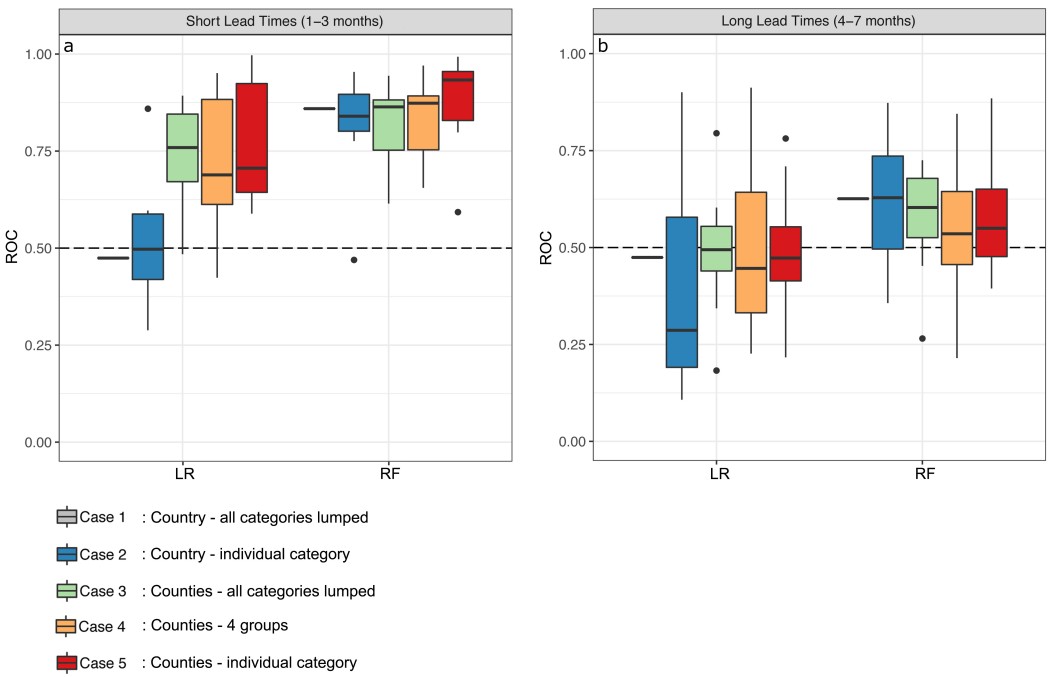

**Figure 4.** Skill of re-forecasted drought impacts for the five Cases using Logistic Regression and Random Forest approaches for the period 2002-2010. The boxes indicate the spread of ROC values for drought impact functions (15 ensemble members) for a) short lead-times (1-3 months), and b) long lead-times (4-7 months). The black dashed line indicates the threshold. All values below this line models have no discriminative capacity to distinguish impact and no impact. The dots stand for models outliers.

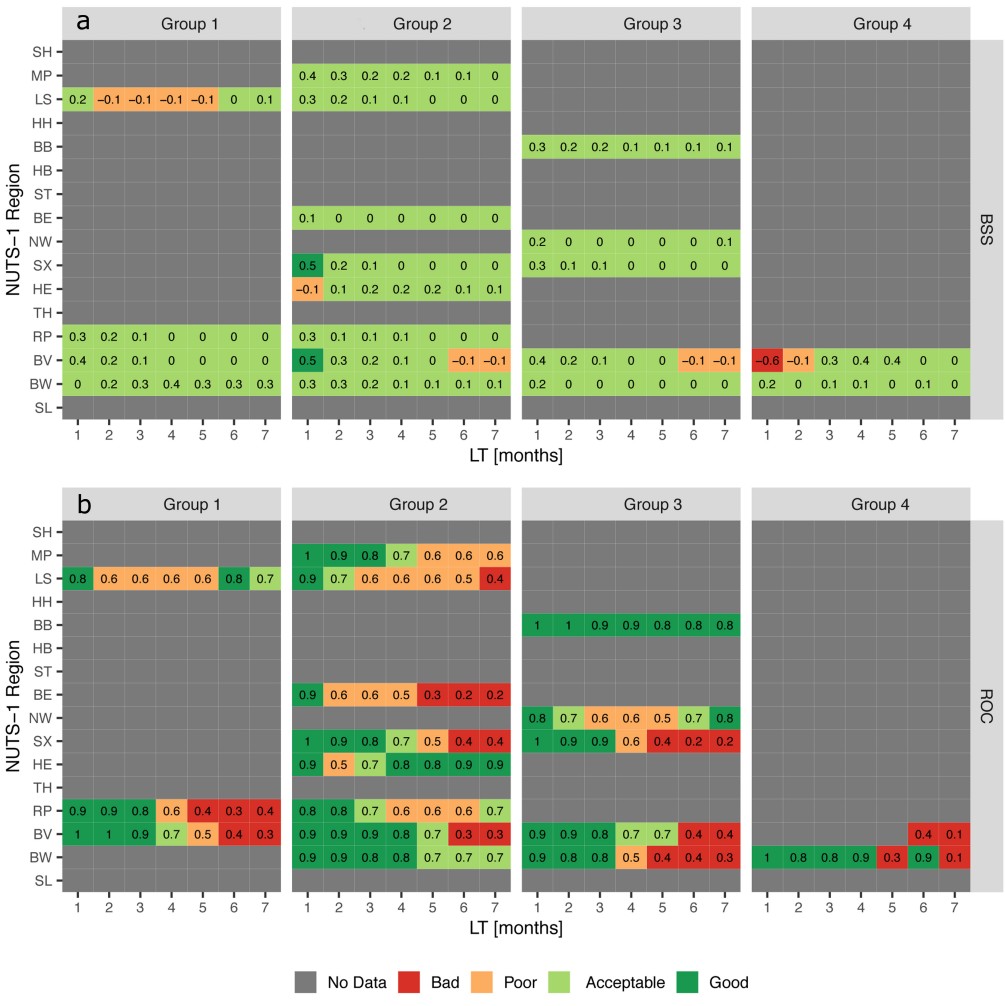

**Figure 5.** Skill of re-forecasted drought impacts for four impact groups and each German NUTS-1 region using the Random Forest approach for the period 2002-2010. The performance is measured using: a) BSS, and b) ROC, for lead times up to 7 months. For the acronym of NUTS-1 regions, see Fig. S1.

**Table 1.** Different Cases for which drought impacts were forecasted. For the German NUTS-1 regions, see Fig. S1

| Case | Spatial resolution | Impact | Ideal number of impact functions per method | Realization number of impact functions per method |
|------|--------------------|--------|---------------------------------------------|---------------------------------------------------|
| 1 | Germany | Impacts of all impacted sectors lumped | 1x1=1 | 1 |
| 2 | Germany | Individual impact category | 1x15=15 | 9 |
| 3 | NUTS-1 level | Impacts of all impacted sectors lumped | 16x1=16 | 12 |
| 4 | NUTS-1 level | Impacted sectors lumped in four groups | 16x4=64 | 19 |
| 5 | NUTS-1 level | Individual impact category | 16x15=240 | 15 |