# Peer review of "Skill of large-scale seasonal drought impact forecasts"

_Natural Hazards and Earth System Sciences, 2020_

## Referee Comment (RC1) · Anonymous Referee #1 · 4 Apr 2020

General Comments: The manuscript submitted showed the efforts devoted to predict drought impacts with lead-times up to 7 months ahead, using the Logistic Regression and Random Forest machine learning approaches. The idea of relating the drought indices to the drought impacts is relatively new and relevant to the journal's scope of understanding the natural hazards and their consequences. However, the machine learning approaches adopted are relatively old-fashioned. It would be nice if the authors can provide better justification for the selected approaches over other methods available. Besides, there are some queries on some statement made by the authors to be justified. Detailed Comments: 1) Abstract: The authors are advised to include more results in the abstract to provide better overview for the readers. 2) Page 1, Line 3: Kindly revise "with a lead-time of 7 months ahead" to "with lead-times up to

7 months ahead" as the study produces predictions with lead-time of 1-,2-,3-,4-,5-,6- and 7-months ahead, not only 7-month. 3) Page 2, Line 40: Kindly revise "Energy and Industry Pubic Water Supply" to "Energy and Industry Public Water Supply" 4) Page 2, Line 40 – 46: The literature reviews show that Logistic Regression (LR) and Random Forest (RF) are already well studied in different studies for deriving the link between drought hazard and their impact. - May I know why are these two methods selected as the approaches in this study? As there are many other approaches available to be further investigated, such as Artificial Neural Network and etc. - Besides, the methods compared have different nature LR (Linear) and RF (Nonlinear). Shouldn't we test the data's linearity before adopting either of these methods? As it will be unfair to the LR (RF) if the nature of the data is nonlinear (linear)? - Recommended recent paper: Drought forecasting: A review of modelling approaches 2007–2017. Journal of Water and Climate Change. 2019 5) Page 2, Line 58: the symbol "box 1" is confusing, kindly revise as "box i" (similar correction for the caption in Figure 1) 6) Page 3, Line 64: Kindly state the full-form of every abbreviation when it is first used, e.g. SRI-x 7) Page 5, Line 153: It is stated that the RF is able to avoid overfitting. To my best knowledge, this statement is wrong as RF does overfit although the generalization error does not increase when the tree size increases. Kindly justify how do the authors avoid overfitting in the current study? How significance is the difference if the cross-validation was adopted? 8) Discussion: The RF showed better performance and the authors claimed that it was due to the longer memory of RF compared to LR. However, the authors never mention about the linearity of the data. Could it be due to the linear/nonlinear nature of the data? Based on the results available, it seems that nonlinear models are favourable, have the authors compare the performance of RF with other nonlinear models? e.g. ANN, Deep learning, and etc. 9) Supporting information, Figure S2: The y-label of histogram for Log Regression is wrong, kindly revise. Besides, may I know how do the authors summarize the predictor importance of few counties into one histogram?

---

## Referee Comment (RC2) · Anonymous Referee #2 · 7 Apr 2020

This study develops drought impact forecasting models based on Logistic Regression and Random Forest in Germany. The hydro-meteorological droughts were quantified using the Standardized Precipitation Index (SPI), the Standardized Precipitation Evaporation Index (SPEI) and the Standardized Runoff Index (SRI) for accumulation periods of 1, 3, 6, and 12 months. The skill of forecasted indices and impacts was evaluated using the Brier Skill Score (BSS) and Relative Operating Characteristic (ROC). The manuscript is well written and structured. The results are sound and discussions (incomplete though) support the results clearly. I have, however, some comments/suggestions that I would like the authors to address.

1- Based on McKee et al. (1993), the length of precipitation record for SPI calculation should be ideally a continuous period of at least 30 years. The same criterion is valid for

[Figure]

SPEI. A short record of 28 years was however used to quantify drought hazards in this study. How would this short record length of the data affect the results? Specifically, how would the results be affected by natural climate variability (laying on oscillation high or low period)?

2- The Standardized Precipitation Index (SPI) and the Standardized Precipitation Evaporation Index (SPEI) were applied to quantify meteorological droughts and the Standardized Runoff Index (SRI) for hydrological droughts, all of them for accumulation periods of 1, 3, 6, and 12 months. While there exist many drought indices, the choice of the SPI, SPEI and SRI indices might be justified. An explanation might also be added on why the authors limited the accumulation periods to 12 months and didn't try longer periods (e.g., 24 months)?

3- There are many methods for calculation of potential evapotranspiration, ranging from simple temperature-based method to the standard Penman-Monteith method. It is not mentioned in the paper which method was used for the potential evapotranspiration calculation for the SPEI index and to simulate gridded runoff?

4- Simulated runoff was used as gridded observed runoff was not available. Is the bias of runoff simulations available to be added to the paper?

5- It was found that the SPEI index shows higher skill than the SPI for short accumulation periods. Can it be because of a longer memory of drought hazard calculated from SPEI which is based on water balance (precipitation minus potential evapotranspiration) compared to SPI which is based on only precipitation?

6- It was found that "drought indices with longer accumulation periods perform better than the ones with short accumulation periods". Isn't it trivial as shorter drought indices with short (long) accumulation periods have more (less) fluctuations/noise and more difficult (easier) to forecast?

Comments on figures:

[Figure]

1- Figures 2-4: The black dashed line in the figures indicates the threshold. It was explained in the caption of Figures 3 and 4, while it was defined as "Thes" in the legend of Figure 2. The same format might be used throughout the paper to keep consistency.

2- Figures 3 and 4: "ROC" can be added as y-axis label in Figures 3 and 4 and removed from the top.

3- Legends of Figures 3 and 4: The legend can be moved to the bottom of the plot with the explanation of each case in the front of it.

4- Caption of Figures 3 and 4: "The performance is measured using ROC" should be removed from the caption of Figure 3 as the plot and the other sentences in the caption clearly show it. Same comment goes for the caption of Figure 4. Revise the sentence in the caption of Figure 4 as "The boxes indicate the spread of ROC values for drought impact functions (15 ensemble members) for a) short lead-times (1-3 months), and b) long lead-times (4-7 months)."

5- Caption of Figure 5: In the statement "four different impact groups" - the word different is not needed. By stating that they are plural, i.e., four impact groups, the logical syntax of the statement means that they must be different.

6- Figure S2: Text on the plots is very small and not readable.

---

## Author Comment (AC1) · 11 May 2020

Reply to Referees

We would like to thank the Referee 1 for the comments, recommendation, and valuable suggestions. In this document, we reply to each of the comments. (Px refers to page number x and Laa-bb refers to line numbers aa to bb in the revised manuscript).

1. General Comments: The manuscript submitted showed the efforts devoted to predict drought impacts with lead-times up to 7 months ahead, using the Logistic Regression and Random Forest machine learning approaches. The idea of relating the drought indices to the drought impacts is relatively new and relevant to the journal's scope of understanding the natural hazards and their consequences.

[Figure]

We would like to thank the referee for the acknowledgment of the novelty of our paper.

2. The machine learning approaches adopted are relatively old-fashioned. It would be nice if the authors can provide better justification for the selected approaches over other methods available. Besides, there are some queries on some statements made by the authors to be justified.

We do agree with the referee that the Logistic Regression (LR) and Random Forest (RF) are not new techniques, but have proven value in some studies that linked drought hazards to impacts (e.g., Blauhut et al., 2015; Stagge et al., 2015; Blauhut et al., 2016; Bachmair et al., 2016; Bachmair et al., 2017). Therefore, we decided to use the LR and RF to forecast drought impacts (previous version P2L38-46). However, those studies reconstructed historical conditions and were not used for drought impact forecasting using dynamical weather forecasts, which is the novelty of our paper. The results are promising and can be implemented in the forecast mode. Additional explanation of why we chose the LR and RF was added in the revised manuscript (P2L54-56 and P3L64-68).

3. Abstract: The authors are advised to include more results in the abstract to provide a better overview for the readers.

We would like to thank the referee for his/her suggestion. More results were added in the abstract (P1L3-15).

4. Page 1, Line 3: Kindly revise "with a lead-time of 7 months ahead" to "with lead-times up to 7 months ahead" as the study produces predictions with lead-time of 1-,2-,3-,4-,5-,6- and 7-months ahead, not only 7-month.

The text was revised (P1L3).

5. Page 2, Line 40: Kindly revise "Energy and Industry, Pubic Water Supply" to "Energy and Industry Public Water Supply".

Unfortunately, we cannot follow up on the suggestion made by the referee to combine

Energy and Industry with Public Water Supply, because they are reported as different impacted sectors in the key paper on the European Drought Impact Inventory, EDII (see Stahl et al., 2016). We would like to keep our study consistent with this.

6.a. Page 2, Line 40 – 46: The literature reviews show that Logistic Regression (LR) and Random Forest (RF) are already well studied in different studies for deriving the link between drought hazard and their impact.

The LR and RF indeed already have been used in previous drought studies (see point 2). However, those studies tried to link the historical drought hazards using the standardized indices, such as SPI and SPEI, to drought impacts using LR and RF. Thus we decided to move one step forward by linking the forecasted drought hazards (SPI, SPEI, and SRI) using dynamical forecasts to drought impacts using the same methods with different combinations of spatial aggregations and impact categories (previous version P2L54-55).

b. May I know why are these two methods selected as the approaches in this study? As there are many other approaches available to be further investigated, such as Artificial Neural Network and etc.

As mentioned above, we selected the LR and RF methods because these were used in previous studies that connected drought hazard to impacts. The use of other methods is foreseen for future study.

c. Besides, the methods compared have different nature LR (Linear) and RF (Nonlinear). Shouldn't we test the data's linearity before adopting either of these methods? As it will be unfair to the LR (RF) if the nature of the data is nonlinear (linear)?

The referee makes a good point here. The input data for the RF and LR are not linear. The drought severity obtained from the standardized indices (i.e. SPI, SPEI, and SRI), i.e. input data, was derived from functions that were developed by fitting the gamma distribution and later were transformed to the normal distribution (previous

version P4L122-125). We added text on this topic in the Discussion as to the main advantage of RF method, which can handle non-linear data better (P10L294-298).

d. Recommended recent paper: Drought forecasting: A review of modeling approaches 2007–2017. Journal of Water and Climate Change. 2019.

The suggested paper was added to the revised manuscript (P10L294).

7. Page 2, Line 58: the symbol "box 1" is confusing, kindly revise as "box i" (similar correction for the caption in Figure 1).

The word on page 2 line 58 writes as box l (BOX L) and not i. We are sorry that we created confusion about the letter. We revised the letters using capital (e.g., A, B, C, and so on) (P2L70-74, and throughout the text).

8. Page 3, Line 64: Kindly state the full-form of every abbreviation when it is first used, e.g. SRI-x.

The full form of the standardized indices was already mentioned in the previous paragraph (e.g., previous version P2L34 for SPI and P2L48 for SPEI). We provided the full-form of SRI in the revised manuscript (P2L58). We thank the reviewer for noticing this.

9. Page 5, Line 153: It is stated that the RF is able to avoid overfitting. To my best knowledge, this statement is wrong as RF does overfit although the generalization error does not increase when the tree size increases. Kindly justify how do the authors avoid overfitting in the current study? How significant is the difference if the cross-validation was adopted?

The referee is correct that the RF is not completely able to avoid overfitting. The RF produces randomly numerous independent trees as an ensemble to reduce the chance of overfitting. The text was revised (P6L170).

One possible way to counteract overfitting is by using cross-validation. In our study, we

did not do the cross-validation (CV). However, we did OOB (out of bag) performance analysis for the development of our RF model, which is not exactly the same but has connections with CV. We think that the calculation of the OOB error in the model training phase is sufficient to test the performance of the model. We added a remark about cross-validation in the revised manuscript (P6L175-178).

10. Discussion: The RF showed better performance and the authors claimed that it was due to the long memory of RF compared to LR. However, the authors never mention about the linearity of the data. Could it be due to the linear/nonlinear nature of the data? Based on the results available, it seems that nonlinear models are favorable, have the authors compare the performance of RF with other nonlinear models? e.g. ANN, Deep learning, and etc.

The referee has a point about the linearity of the data. Drought indices used as input in the machine learning models are not linear (see point 6 above). We discussed the non-linearity of our data in the revised manuscript (P10L294-298). We did not compare the results with other machine learning models. However, some previous studies concluded that RF produces better performance compared to other Machine Learning approaches (e.g., Boosted regression trees, cubist, decision trees, Hurdle, and logistic regression; Park et al., 2016; Rhee and Im, 2017; Bachmair et al., 2017) (previous version P9L264-266).

11. Supporting information, Figure S2: The y-label of the histogram for Log Regression is wrong, kindly revise. Besides, may I know how do the authors summarize the predictor importance of few counties into one histogram?

We thank the referee for his/her careful reading of our manuscript. The figure was revised accordingly. We plotted the histograms based on the average of the predictor of importance for each county. An explanation was added to the revised Supplementary Material.

References Bachmair, S., Svensson, C., Prosdocimi, I., Hannaford, J., &

Stahl, K. (2017). Developing drought impact functions for drought risk management. Natural Hazards and Earth System Sciences, 17 (11), 1947–1960. https://doi.org/10.5194/nhess-17-1947-2017.

Bachmair, S., Svensson, C., Hannaford, J., Barker, L. J., & Stahl, K. (2016). A quantitative analysis to objectively appraise drought indicators and model drought impacts. Hydrol. Earth Syst. Sci., 20, 2589–2609, https://doi.org/10.5194/ hess-20-2589-2016.

Blauhut, V., Stahl, K., Stagge, J. H., Tallaksen, L. M., De Stefano, L., & Vogt, J. (2016). Estimating drought risk across Europe from reported drought impacts, drought indices, and vulnerability factors. Hydrol. Earth Syst. Sci., 20, 2779-2800, doi:10.5194/hess-20-2779-2016.

Blauhut, V., Gudmundsson, L., & Stahl, K. (2015). Towards pan-European drought risk maps: quantifying the link between drought indices and reported drought impacts. Environmental Research Letters, 10 (1), 014008, https://doi.org/10.1088/1748-9326/10/1/014008.

Park, S., Im, J., Jang, E., & Rhee, J. (2015). Drought assessment and monitoring through blending of multi-sensor indices using machine learning approaches for different climate regions. Agric. For. Meteorol., 216, 157–169, https://doi.org/10.1016/j.agrformet.2015.10.011.

Rhee, J., & Im, J. (2017). Meteorological drought forecasting for ungauged areas based on machine learning: Using long-range climate forecast and remote sensing data. Agric. For. Meteorol., 237-238, 105–122, https://doi.org/10.1016/j.agrformet.2017.02.011.

Stagge, J. H., Kohn, I., Tallaksen, L. M., & Stahl, K. (2015). Modeling drought impact occurrence based on meteorological drought indices in Europe. Journal of Hydrology, 530, 37–50, http://dx.doi.org/10.1016/j.jhydrol.2015.09.039.

Stahl, K., Kohn, I., Blauhut, V., Urquijo, J., De Stefano, L., AcaÌ́Acio, V., Dias, S.,

Stagge, J. H., Tallaksen, L. M., Kampragou, E.,......Van Lanen, H. A. J. (2016). Impacts of european drought events: insights from an international database of text-based reports. Natural Hazards and Earth System Sciences, 16 (3), 801–819, https://doi.org/10.5194/nhess-16-801-2016.

---

## Author Comment (AC2) · 11 May 2020

Reply to Referees

We would like to thank the Referee 2 for the comments, recommendation, and valuable suggestions. In this document, we reply to each of the comments. (Laa-bb refers to line numbers aa to bb and Px refers to page number x in the revised manuscript).

1. The manuscript is well written and structured. The results are sound and discussions (incomplete though) support the results clearly. I have, however, some comments/suggestions that I would like the authors to address.

We thank the referee for his/her positive response and support in improving our manuscript.

[Figure]

2. Based on McKee et al. (1993), the length of precipitation record for SPI calculation should be ideally a continuous period of at least 30 years. The same criterion is valid for SPEI. A short record of 28 years was however used to quantify drought hazards in this study. How would this short record length of the data affect the results? Specifically, how would the results be affected by natural climate variability (laying on oscillation high or low period)?

The referee has a point here that indeed the standardized indices require preferably 30 years record. However, we could only use 28 years of observational record (proxy) from 1990 to 2017 due to data availability. EFAS data before 1990 are not available and we only had data up to 2017. The length of the observational data might have some implications on the calculation of parameters of the monthly distributions, which affects the calculation of the drought severity index. For example, if the records (< 30 years) do not include extreme low/high events, then the calculation of drought indices will overestimate these low/high severities, if these would occur, due to lack of outliers in the low/high of normal distribution. For our study, we argue that 2 years missing in the record do not significantly influence our results for following reasons: 1) years 1988 and 1989 were not recorded as extreme drought years, 2) we included drought years in parts of Europe, e.g. 1991-1992, 1995, 1996-1997, 2003, 2006, 2008, and 2015 (Spinoni et al., 2015). We added information about the influence of data length in the revised manuscript (P9L266-274).

3. The Standardized Precipitation Index (SPI) and the Standardized Precipitation Evaporation Index (SPEI) were applied to quantify meteorological droughts and the Standardized Runoff Index (SRI) for hydrological droughts, all of them for accumulation periods of 1, 3, 6, and 12 months. While there exist many drought indices, the choice of the SPI, SPEI, and SRI indices might be justified. An explanation might also be added to why the authors limited the accumulation periods to 12 months and didn't try longer periods (e.g., 24 months)?

The SPI, SPEI, and SRI were chosen in our study, because first these indices are

widely used both in the scientific community and by end-users, such as water agencies. Second, these indices were used in previous studies to link the historical drought hazards with its impacts and showed promising results to develop the drought impact functions (previous version P2L47-52). We decided to use the standardized indices with accumulation periods of 1, 3, 6, and 12 months because higher accumulation periods, e.g. beyond 12 months, consist of long series of observation data relative to the lead time, which is a caveat for forecasting skill assessment. Higher proportion of observed data will artificially inflate forecast scores (Sutanto et al., 2020). For example, SPI-12 with a lead-time of 1-month consists of 11 months observed data and 1-month forecast data. SPI-12 with a lead-time of 7-month consists of 5 months of observed data and 7 months forecasts. We added this information in the revised manuscript (P11L337-342).

4. There are many methods for calculation of potential evapotranspiration, ranging from simple temperature-based methods to the standard Penman-Monteith method. It is not mentioned in the paper which method was used for the potential evapotranspiration calculation for the SPEI index and to simulate gridded runoff?

Potential evapotranspiration is calculated through the offline LISVAP pre-processor based on the Penman-Monteith equation (Van Der Knijff, 2008). This information was added to the revised manuscript, including the reference (P4L119-120).

5. Simulated runoff was used as gridded observed runoff was not available. Is the bias of runoff simulations available to be added to the paper?

Observed runoff data are not available, these cannot be measured by definition. Streamflow can be measured, but runoff cannot. Thus, we could not calculate the bias of simulated runoff. Streamflow data in some rivers are available (>250 catchments across Europe) and these were used in previous studies to calibrate and validate the hydrological model, LISFLOOD, from which outcome has been applied in this study (e.g., Feyen and Dankers, 2009; Forzieri et al. 2014). LISFLOOD obtained a me-

dian Nash Sutcliffe Efficiency (NSE) of 0.57 over the validation period. We added this information in the revised manuscript (P4L122-125).

6. It was found that the SPEI index shows higher skill than the SPI for short accumulation periods. Can it be because of a long memory of drought hazard calculated from SPEI which is based on water balance (precipitation minus potential evapotranspiration) compared to SPI which is based on only precipitation?

The SPEI drought forecasts have slightly higher skill than the SPI (Fig. 2), especially in autumn and winter. In general, the source of predictability of SPEI comes from the higher temperature predictability due to NAO in Europe than precipitation. Temperature is one of the weather variables that controls potential evapotranspiration. We added this information in the revised manuscript (P7L212-214). Longer memory of drought hazard is only found for the hydrological drought (here runoff, SRI approach), associated with memory represented in initial hydrological conditions and storage in the hydrological system (previous version P10L302).

7. It was found that "drought indices with longer accumulation periods perform better than the ones with short accumulation periods". Isn't it trivial as shorter drought indices with short (long) accumulation periods have more (less) fluctuations/noise and more difficult (easier) to forecast?

The score of meteorological drought forecasts improves with the increase of the accumulation periods of the SPI and SPEI because of a higher proportion of observed data, which artificially inflates forecast scores (Sutanto et al., 2020) (see point 3). Likely, forecast skill of drought indices with longer accumulation periods (long drought indices) also is positively affected by the lower fluctuation/noise in the index compared to shorter accumulation periods. However, we cannot attribute the higher skill of the longer drought indices either to longer period of observations in the forecast, or to the smoother time series. We added this information in the revised manuscript (P11L337-342).

8. Figures 2-4: The black dashed line in the figures indicates the threshold. It was explained in the caption of Figures 3 and 4, while it was defined as "Thres" in the legend of Figure 2. The same format might be used throughout the paper to keep consistency.

We agree with the referee. We revised Figure 2 accordingly (P19).

9. Figures 3 and 4: "ROC" can be added as y-axis label in Figures 3 and 4 and removed from the top.

We moved the ROC legend to the y-axis in Figures 3 and 4 in the revised manuscript (P20-21).

10. Legends of Figures 3 and 4: The legend can be moved to the bottom of the plot with the explanation of each case in front of it.

We moved the legend in Figures 3 and 4 accordingly (P20-21).

11. Caption of Figures 3 and 4: "The performance is measured using ROC" should be removed from the caption of Figure 3 as the plot and the other sentences in the caption clearly show it. The same comment goes for the caption of Figure 4. Revise the sentence in the caption of Figure 4 as "The boxes indicate the spread of ROC values for drought impact functions (15 ensemble members) for a) short lead-times (1-3 months), and b) long lead-times (4-7 months)."

It is a good suggestion. We revised Figure captions accordingly (P20-21).

12. Caption of Figure 5: In the statement "four different impact groups" - the word different is not needed. By stating that they are plural, i.e., four impact groups, the logical syntax of the statement means that they must be different.

We removed the word different in the caption (P22).

13. Figure S2: Text on the plots is very small and not readable.

We revised the Figure by placing figures S2a (now S2) at the top and S2b (now S3) at the bottom to increase the readability (SM P2-3).

References Spinoni, J., G. Naumann, J. V. Vogt, and P. Babosa., 2015: The biggest drought events in Europe from 1950 to 2012. Journal of Hydrology: Regional Studies, 3, 509-524, http://dx.doi.org/10.1016/j.ejrh.2015.01.001.

Sutanto, S. J., F. Wetterhall, and H. A. J. Van Lanen., 2020: Hydrological drought forecasts outperform meteorological drought forecasts. Environ. Res. Lett. in press https://doi.org/10.1088/1748-9326/ab8b13.

Van Der Knijff, J. 2008. LISVAP: Evaporation Pre-Processor for the LISFLOOD Water Balance and Flood Simulation Model. EUR 22639 EN/2.

Feyen, L., and R. Dankers, 2009: Impact of global warming on streamflow drought in europe. J. Geophys. Res., 114, D17 116, doi:10.1029/2008JD011438.

Forzieri, G., L. Feyen, R. Rojas, M. Flörke, F. Wimmer, and A. Bianchi., 2014: Ensemble projections of future streamflow droughts in europe. Hydrol. Earth Syst. Sci., 18, 85–108, doi: 10.5194/hess-18-85-2014.